# Mechanisms Involved in Epileptogenesis in Alzheimer’s Disease and Their Therapeutic Implications

**DOI:** 10.3390/ijms23084307

**Published:** 2022-04-13

**Authors:** Miren Altuna, Gonzalo Olmedo-Saura, María Carmona-Iragui, Juan Fortea

**Affiliations:** 1Sant Pau Memory Unit, Department of Neurology, Hospital de la Santa Creu i Sant Pau, Biomedical Research Institute Sant Pau, Universitat Autònoma de Barcelona, 08041 Barcelona, Spain; golmedo@santpau.cat (G.O.-S.); mcarmonai@santpau.cat (M.C.-I.); jfortea@santpau.cat (J.F.); 2Center of Biomedical Investigation Network for Neurodegenerative Diseases (CIBERNED), 28031 Madrid, Spain; 3CITA-Alzheimer, 20009 Donostia-San Sebastián, Spain; 4Barcelona Down Medical Center, Fundació Catalana de Síndrome de Down, 08029 Barcelona, Spain

**Keywords:** seizures, epilepsy, Alzheimer’s disease, antiseizure medications, hyperexcitability

## Abstract

Epilepsy and Alzheimer’s disease (AD) incidence increases with age. There are reciprocal relationships between epilepsy and AD. Epilepsy is a risk factor for AD and, in turn, AD is an independent risk factor for developing epilepsy in old age, and abnormal AD biomarkers in PET and/or CSF are frequently found in late-onset epilepsies of unknown etiology. Accordingly, epilepsy and AD share pathophysiological processes, including neuronal hyperexcitability and an early excitatory–inhibitory dysregulation, leading to dysfunction in the inhibitory GABAergic and excitatory glutamatergic systems. Moreover, both β-amyloid and tau protein aggregates, the anatomopathological hallmarks of AD, have proepileptic effects. Finally, these aggregates have been found in the resection material of refractory temporal lobe epilepsies, suggesting that epilepsy leads to amyloid and tau aggregates. Some epileptic syndromes, such as medial temporal lobe epilepsy, share structural and functional neuroimaging findings with AD, leading to overlapping symptomatology, such as episodic memory deficits and toxic synergistic effects. In this respect, the existence of epileptiform activity and electroclinical seizures in AD appears to accelerate the progression of cognitive decline, and the presence of cognitive decline is much more prevalent in epileptic patients than in elderly patients without epilepsy. Notwithstanding their clinical significance, the diagnosis of clinical seizures in AD is a challenge. Most are focal and manifest with an altered level of consciousness without motor symptoms, and are often interpreted as cognitive fluctuations. Finally, despite the frequent association of epilepsy and AD dementia, there is a lack of clinical trials to guide the use of antiseizure medications (ASMs). There is also a potential role for ASMs to be used as disease-modifying drugs in AD.

## 1. Introduction

Alzheimer’s disease (AD) is the most common neurodegenerative disorder and the leading cause of dementia, accounting for over 60–70% of dementia cases [1]. Epilepsy is the third most frequent neurological condition in the elderly after cerebrovascular pathology and neurodegenerative dementias [2]. AD dementia and epilepsy frequently coexist, and there are reciprocal relationships between the two diseases. AD is an independent risk factor for epilepsy with an increased risk ranging from 2 to 10 times compared to age-matched healthy controls [3,4,5,6,7,8,9,10,11,12,13,14,15,16]. Conversely, epilepsy, especially late-onset epilepsy of unknown etiology (no cause identified after completing etiological study), has been described as a risk factor for the development of AD [9,13,17,18,19,20]. In addition, AD and epilepsy share several risk factors, for instance, cardiovascular risk factors; blood–brain barrier dysfunction; cerebrovascular damage (both micro- and macrovasculature); a personal history of brain traumatic injury; the presence of the ε4 allele of the APOE gene; and, most importantly, advanced age [17,21]. Importantly, both β-amyloid and tau protein aggregates have proepileptic effects [13,22,23,24]. Convergently, in surgical material from refractory mesial temporal lobe epilepsies, the presence of amyloid and tau proteins has been detected [25].

There are shared pathophysiological processes both in AD and epilepsy. Both present a dysregulation of the excitatory–inhibitory tone, which is presumably caused by alterations in the glutamatergic (excitatory) and GABAergic (inhibitory) systems. AD and epilepsy, especially some epileptic syndromes, such as medial temporal lobe epilepsy, predominantly target similar brain regions (CA1, subiculum, and entorhinal cortex) [26,27], leading to overlapping symptoms, such as episodic memory deficits and alterations in similar large-scale networks, especially the default neural network [26,27]. Therefore, a detailed electroclinical characterization of both AD-associated epilepsy and late-onset epilepsy of unknown etiology (LOEU) may facilitate early diagnoses.

Our objectives are to review the epidemiology, etiopathogenic mechanisms, and risk factors, as well as the clinical overlap between symptomatic AD and epilepsy. We also review available evidence to guide the use of antiseizure medications (ASMs) in AD and the rationale of ASMs being potentially AD-disease-modifying treatments.

## 2. Methodology

We performed a literature review on 24 January 2022 using PubMed and Web of Science (WOS), combining the Mesh Terms “Alzheimer Disease”, “Epilepsy”, “Seizures”, and “Anticonvulsants” (Figure 1). We did not apply any time restriction, and we included original articles and review articles with data from human subjects (exclusion of papers in animals only) written in English, Spanish, or French. We selected articles with abstracts available in PubMed or WOS. After reading the titles and abstracts, papers that met eligibility criteria were selected for full-text revision. Papers specifically dealing with epilepsy and Alzheimer’s disease and the potential benefits of antiseizure medications beyond their antiepileptic effect were included in this review.

## 3. Epilepsy in Alzheimer’s Disease

The prevalence of epilepsy in sporadic preclinical, prodromal, and AD dementia is higher than in healthy age-matched controls [11,14,28,29]. Genetically determined AD (autosomal dominant AD (ADAD) and Down syndrome-associated AD (DSAD)) is at particularly high risk: Down syndrome > *APP* > *PSEN2* > *PSEN1* mutations [13,29,30,31,32,33,34]. Interestingly, in agreement with the ultrahigh risk in DSAD, ADAD patients with amyloid precursor protein (*APP*) gene duplications (57%) have a higher risk than those with presenilin 1 (*PSEN1*) (37%) and presenilin 2 (*PSEN2*) mutations (31%) [29,35,36,37,38]. In the specific case of *PSEN1*, mutations that occur before codon 200 have been associated with a higher risk for epilepsy [39,40]. In subjects with Down syndrome and AD, the risk of developing epilepsy ranges from 46 to 84%, and a specific type of epilepsy called late-onset myoclonic epilepsy has been described [30,33].

Table 1 summarizes the sociodemographic and clinical risk factors for the development of epilepsy in AD, as well as the comorbidities increasing the risk of epilepsy [4,6,9,12,22,26,41,42,43,44,45,46,47,48,49,50,51,52,53,54]. It is noteworthy that although the risk of epileptic seizures seems to increase with disease severity, the risk is already increased in the prodromal and preclinical stages of the disease [20,31].

The semiology of seizures varies among the different forms of AD. In sporadic AD, the most frequent type is focal epileptic seizures, presenting with an altered level of consciousness without motor symptoms (55–70%) [6,14,47,51,55,56,57]. The most frequent semiology of these nonmotor focal seizures corresponds to self-limited episodes of amnestic spells, aphasia of expression or comprehension or mixed, *déjà vu* or *jamais vu*, sensory phenomena (positive or negative), staring spells, and unexplained emotions. All this symptomatology is often erroneously interpreted as cognitive fluctuations, which are frequent in sporadic AD [51,58]. In ADAD, the seizure semiology is more varied and more frequently has a motor component in the form of focal seizures and/or bilateral tonic–clonic and myoclonic seizures [32,38,40,43,53,54,59]. Finally, in DSAD, the most frequent seizures are bilateral tonic–clonic and myoclonic seizures [30,33].

The identification of epilepsy in AD is, therefore, difficult [51,58]. In this context, a routine electroencephalogram (EEG) at symptomatic AD diagnosis might be advisable, especially in genetically determined AD. The EEG should, nonetheless, be interpreted as a supportive diagnostic tool, always assessed in the clinical context of the patient (assigning more value to the semiology of the seizures than to the findings of a specific recording), as the absence of interictal epileptiform discharges (IEDs) does not exclude epileptiform etiology, and the existence of IEDs does not necessarily imply that the patient has epileptic seizures. Multiple nonepileptiform, unspecific (diffuse slowing and continuous generalized periodic discharges with triphasic morphology), and ictal and interictal, epileptiform abnormalities (sharp waves, spikes, spike-waves, and polyspike-waves) are detected in the surface EEG of subjects with AD with or without previous epilepsy diagnosis [9,60]. Indeed, the presence of IEDs is more frequent (up to 4 times) in subjects with AD with respect to age-matched healthy controls [46,51,56,61,62,63]. The presence of IEDs, however, is associated with a higher risk for clinical seizures. In subjects with AD and epilepsy, IEDs are twice as frequent as in AD dementia patients without epilepsy. The diagnostic yield is, however, suboptimal [6,14,16,23,36,45,64,65,66]. The low diagnostic performance of surface EEG seems to be related to the focal character and preferential temporal localization (with lower representation in surface EEG) of IEDs in AD [13,22,51]. In this respect, some IEDs confer more risk. High-frequency IEDs of the right temporal location during wakefulness and during REM sleep are associated with a higher risk of developing seizures [65]. Another factor contributing to the lower diagnostic performance is the fact that an EEG is routinely performed during wakefulness. IEDs are more frequent during sleep, especially in the N2 phases [14,31,51,60,62,67]. Finally, patients with AD with or without epilepsy frequently show EEG rhythm abnormalities. In this respect, the use of quantitative EEG in AD has also shown an increase in delta and theta frequency ranges and a decrease in alpha and beta power with respect to controls [9,60].

Epilepsy in the context of AD has an impact on both AD biology and clinical course [68]. The presence of electroclinical seizures seems to accelerate the progression of cognitive multidomain impairment (memory, executive, and visuospatial functions) and may contribute to the more rapid loss of functional autonomy [4,15,36,62,63,67,69,70,71,72,73,74]. A similar impact of IEDs in the absence of observed clinical seizures in AD has also been reported in recent years [67,74], especially if IEDs occur in the left temporal location [74] and during the slow-wave sleep phase [65], a brain region and a sleep phase that are essential for memory consolidation, respectively.

### 3.1. Late-Onset Epilepsy

The risk of developing epilepsy increases with age, being 2 to 6 times higher after 55 years of age than in young adults [18]. Late-onset epilepsy is defined when the onset occurs after 55 years of age [18,20,75]. The risk, nevertheless, continues to increase even after this age, with a reported incidence of untriggered seizures of 80.8 cases/100.000 inhabitants/year at age 60 compared to 135–175 cases/100.000 inhabitants/year at age 80 [76,77,78].

Multiple causes of late-onset epilepsy have been described (Figure 2). The majority of them are related to acquired cerebral damage, most frequently cerebral vascular damage [8,20,79,80]. However, in recent years, there has been an increasing appreciation of the importance of neurodegenerative diseases, especially AD [20,81], as the underlying cause. It is important to note that despite the improvement in diagnostic tools, up to 20–33% of cases remain unknown (late-onset epilepsy of unknown etiology or LOEU) [2,8,20,82,83]. Men are at a higher risk for late-onset epilepsy, LOEU, and the epilepsy associated with AD [20]. Another risk factor for late-onset epilepsy is the ε4 allele of APOE, which, in turn, is the major genetic risk factor for sporadic AD [18,80].

In late-onset epilepsies, the semiology of seizures is very variable and is related to the location of the underlying brain damage origin of the ictal activity. The most frequent clinical seizures (in approximately 66%) are focal, with an altered level of consciousness without motor symptoms [2,8]. The low frequency of motor symptoms and the high frequency of an altered level of consciousness in late-onset epilepsy make the early identification of these episodes difficult [8]. The EEG is not sensitive enough to address this diagnostic challenge. Only 29% of routine EEGs in subjects with late-onset epilepsy have IEDs [83].

Cognitive impairment is a common finding in epilepsy, including late-onset epilepsy [20,84]. Indeed, multiple studies support a higher prevalence of mild cognitive impairment (MCI) in late-onset epilepsy (40–55%) [17,75] and in temporal lobe epilepsies (TLE) in which the prevalence is up 60% [84]. The numbers of temporomedial IEDs and hippocampal onset seizures correlate with the progressive episodic memory decline in TLE [20,78,85]. In turn, up to 50% of LOEU patients have frequent multidomain, dysexecutive-predominant MCI with the frequent involvement of visuospatial functions [13,86]. Whether the MCI described in the context of LOEU is a consequence of the presence of electroclinical seizures and IEDs or instead reflects the existence of a previously undiagnosed neurodegenerative dementia is not yet fully resolved, and the two options are not mutually exclusive. Changes in cerebrospinal fluid (CSF) AD biomarkers, particularly reductions in Aβ1-42, have been recently reported in LOEU patients, mainly but not exclusively in patients with MCI [22]. In the same line, progression to dementia in LOEU patients might be as high as 22% after 10 years of follow-up, especially in those with MCI and reduced Aβ1-42 levels at LOEU diagnosis [22].

### 3.2. Epileptogenic Mechanisms in Alzheimer’s Disease

Multiple mechanisms involved in the increased risk of epilepsy in AD have been described. These mechanisms are linked to neurotransmitters involved in the excitatory–inhibitory balance (glutamate, GABA, acetylcholine, and noradrenaline), alterations in ion channels (sodium, potassium, and calcium channels), changes in neuronal networks, anatomopathological hallmarks of AD (amyloid and tau), neuroinflammation, and genetic risk factors. All these mechanisms lead to modifications of synaptic integrity and activity reflected by changes in long-term potentiation (LTP) and long-term depression (LTD) and induce a state of neuronal hyperexcitability.

Neuronal hyperexcitability is a physiological phenomenon associated with aging but is clearly exacerbated in some neurological diseases, such as AD [45,67]. This neuronal hyperexcitability in AD starts in the dentate gyrus, spreads to the hippocampus, and finally affects the rest of the brain [23,38].

Excitatory–inhibitory imbalance:

Normal brain function requires the existence of an excitatory–inhibitory balance and synapse homeostasis. Minimal changes in these processes increase the probability of epileptiform activity, electroclinical seizures, and cognitive impairment [23,45,87] and, in AD, both these processes are affected [23,45].

#### 3.2.1. Role of Neurotransmitters in Epileptogenesis

The main excitatory neurotransmitter in the central nervous system (CNS) is glutamate, and the main inhibitory neurotransmitter is GABA. Both are altered in AD and epilepsy [23,45].

The increased glutamatergic tone has been linked to glutamate–glutamine cycle disturbances, which lead to increased extracellular glutamate and decreased GABA levels [36,87,88]. In AD animal models, glutamatergic N-methyl-D-aspartate receptor (NMDAR) activation increases beta-secretase activity, and promotes the formation of amyloid plaques [89,90,91], tau hyperphosphorylation [15], and cell death [87]. GABAergic dysfunction has gained increasing attention in recent years. In this respect, a significant reduction in GABA concentration in the temporal lobe, the selective reduction in GABAergic inhibitory interneurons, and a reduction in GABAergic terminals, especially in the areas closest to amyloid plaques, have all been reported in animal models [17,23,92,93]. This decreased GABAergic tone [16] leads to abnormal cortical hypersynchronization and could also decrease neuro- and synaptogenesis [92,93].

Other neurotransmitters, such as acetylcholine and noradrenaline in epilepsy and AD, also influence neuronal hyperexcitability, but their role seems to be of lesser magnitude than those of glutamate and GABA. A compensatory increase in cholinergic tone in relation to neurodegeneration in the nucleus basalis of Meynert has been linked to the neuronal hyperexcitability and subclinical epileptiform activity in AD animal models [15,19,36]. Noradrenaline has antiepileptic effects in animal models. The early degeneration of noradrenergic neurons in the locus ceruleus in AD impedes the compensatory increase in noradrenaline levels in the hyperexcited hippocampus [19,36].

#### 3.2.2. Ion Channel Disruptions

AD impacts the number and function of voltage-dependent sodium (Na^+^), calcium (Ca^+^) and potassium (K^+^) ion channels [15]. These ion channels contribute both to the generation and maintenance of epileptic seizures and AD pathophysiology as they can also increase glutamate-mediated excitotoxicity and neuronal hyperexcitability [89,90,91]. Intracellular calcium, in particular, must be closely regulated for the maintenance of the excitatory–inhibitory balance. An impairment of intracellular calcium regulation has been reported in both AD and epilepsy [38,40]. Interestingly, beta-secretase 1 (BACE 1), which is hyperrepressed in AD, is able to modulate the expression and the functionality of voltage-dependent potassium channels [17,19,22,94]. In addition, in animal models of AD, an increased level of L-type calcium channels (essential for synchronous calcium oscillations), the overexpression of voltage-dependent sodium channel Nav 1.6 [22], and a decrease in the sodium channel Nav 1.1 levels in GABAergic interneurons, all of which have been previously related to different epileptic syndromes, have been related to increased hyperexcitability in the context of AD [91,92,94]. In turn, the altered expression and/or function of voltage-dependent ion channels, and a reduction in the calbindin protein responsible for intracellular calcium transport in the dentate gyrus have also been demonstrated in animal models of AD, and this reduced expression has also been associated with a reduced seizure threshold and increased difficulties in memory consolidation [22,95].

#### 3.2.3. Network Dysfunction

Neural network dysfunction plays an important role in AD [34,39,96]. This network dysfunction can be assessed using functional MRI and/or quantitative EEG [34]; it is present decades before the onset of symptoms [24] and has been linked to cognitive deficits. Normal neuronal synchrony, which is closely linked to neuronal network integrity, is essential for the creation of oscillatory brain rhythms, and the rhythmic fluctuations of electrical activity. In turn, this neuronal synchrony is the basis of various cognitive functions, including memory [24]. AD is associated with an early disruption of gamma oscillations [51,60], synaptic dysfunction, and synaptic depression [24].

### 3.3. Amyloid and Tau Promote Hyperexcitability and Facilitate Epileptogenesis

There is growing evidence of the potential proepileptic role of the anatomopathological hallmarks of AD (amyloid and tau) [4,39].

#### 3.3.1. Amyloid (Aβ)

Aβ, which begins to accumulate 20 years before symptom onset, has been linked to hyperexcitability [39], synaptic dysfunction, and neuronal death [67]. Fibrillar or oligomeric forms of Aβ (pre-plaques stages) contribute to a larger extent than amyloid plaques to neuronal hyperexcitability in cortical and hippocampal neurons, the change in slow-wave oscillations, and the increase in IEDs and network hypersynchrony [13,23,28,36,45,88,97].

The mechanisms through which soluble forms of Aβ exert these effects include an increase in the glutamatergic [22,38,87] tone, a reduction in GABAergic activity [36,53,87], a dysregulation of the activity of voltage-dependent ion channels (which induce spontaneous action potentials [22]) and an increase in proinflammatory cytokines [87] (Figure 3), which also favor epileptogenicity. It is of note that neuronal hyperexcitability increases the deposition and propagation of amyloid and tau proteins, leading to a feedforward cycle [13,28,31,32,39,45,67]. In this respect, IEDs, even in the absence of anatomopathological hallmarks of AD, can promote Aβ deposition. Childhood-onset epilepsy (before the age of 5 years) is associated with increased amyloid PET uptake in the sixth decade of life when compared to age-matched controls [92].

#### 3.3.2. Tau

The tau protein regulates the stability and dynamics of the cytoskeleton of neurons. The phosphorylation of tau is necessary for its correct functioning. However, its hyperphosphorylation can induce its dysfunction with the loss of stability of the cytoskeleton, potentially damaging axonal transport, inducing synaptic loss and finally neuronal death [13]. Intracellular deposits of hyperphosphorylated tau have been found in brain traumatic injury, epilepsy (resection samples of refractory temporal lobe epilepsy), and AD [12,92].

In turn, in both AD and epilepsy, the dysfunction of the tyrosine kinase Fyn has also been described, which has been attributed a role in neuronal hyperexcitability through the modulation of both glutamatergic and GABAergic receptors and its association with different ion channels with both excitatory and inhibitory functions [98]. In addition to its involvement in transmission and synaptic plasticity, it is also linked to the development of dendritic spines and to the process of tau protein phosphorylation and its subsequent intracellular deposition in the form of neurofibrillary tangles in AD [99].

Neurofibrillary tangles at autopsy load have been associated with neuronal hyperexcitability and risk of epilepsy [15,36,39,55,100]. Similarly, patients with AD and epilepsy have higher CSF tau levels than those without epilepsy [36,100]. Finally, soluble forms of tau, prior to the formation of neurofibrillary tangles, are also able to increase the glutamatergic tone and to induce neuronal network reorganization, increasing the amount of IEDs and electroclinical seizures in animal models of AD [15,17,19,22,51,56,88] (Figure 3).

### 3.4. Other Mechanisms

#### 3.4.1. Neuroinflammation

Neuroinflammation occurs early in AD, even before the deposition of amyloid plaques. It is also present in some types of epilepsies, including TLE [22,101]. Both in AD and epilepsy, astrogliosis and microgliosis (activation and proliferation of astrocytes and microglia) alter the glutamate–glutamine cycle. In particular, astrocytes and glial cells secrete proinflammatory cytokines (IL-6, IL-1β, and TNF-α) that can modulate the release of glutamate and modify its postsynaptic reuptake [22,26,36,87]. They also reduce GABA signaling [22].

Blood–brain barrier (BBB) dysfunction occurs in both AD [102] and TLE and is thought to be, at least in part, a consequence of the proinflammatory response (astrogliosis and release of proinflammatory cytokines) [103]. However, at the same time, the loss of barrier integrity favors the perpetuation of a proinflammatory environment (increase in toxic substances in the CNS, which, in turn, leads to the activation of microglia and the release of proinflammatory cytokines again) and, with it, also a dysregulation of the excitatory–inhibitory system and also pro-excitatory synaptic dysfunction [66]. This excitatory–inhibitory disbalance could increase the probability of the existence of interictal epileptiform activity and also of clinical and/or electrical epileptic seizures [102,103]. It has also been postulated that blood–brain barrier dysfunction could increase Aβ production through the stimulation of β and γ-secretases, and the pro-excitatory effect of Aβ is well known [104].

#### 3.4.2. mTOR

mTOR (mammalian target of rapamycin) is a serine/threonine kinase expressed in multiple cell types and involved in the regulation of essential cell functions, such as proliferation or transcription [105]. From early preclinical stages of AD, it is involved in the generation of Aβ42 and its washout, tau protein synthesis, and endoplasmic reticulum stress [91]. In cell cultures and animal models of AD, it has been shown that Aβ and GSK3β (Glycogen synthase kinase 3β), an enzyme involved in the hyperphosphorylation of the tau protein, also activates mTOR [105]. Activated mTOR is thought to reduce the autophagy capacity necessary to eliminate neurotoxic substances such as Aβ and phosphorylated tau, and this accumulation of toxic substances would contribute to neuronal death [105]. Both in TLE and AD, an hyperactivation of mTOR has been reported [17].

#### 3.4.3. Apolipoprotein (APOE)

The APOE ε4 allele is the most important genetic risk factor for sporadic AD, and is also a risk factor for post-traumatic epilepsy [26,106]. In AD, the APOE ε4 allele is related to the increased impairment of and reduction in GABAergic interneurons, and more severe damage to BBB integrity [107], and is believed to favor a decreased inhibitory tone [26,38,53]. In addition, the APOE ε4 haplotype can also influence the clinical phenotype of TLE. Carriers have an earlier onset of seizures, increased risk of postictal confusion, longer standing seizures, lower probability of seizure control with ASMs, and higher verbal and memory deficits [19]. It is of note that in cognitively normal adults, APOE ε4 allele carriers have more frequent IEDs (sharp waves in hyperventilation) compared to noncarriers [26,53].

## 4. Antiseizure Medications (ASMs) in Alzheimer’s Disease

### 4.1. Treating Epileptic Seizures in AD

In the context of symptomatic AD, there is a 70% risk of seizure recurrence after a first episode. Therefore, indefinite treatment with ASMs is advisable after a first untriggered seizure [4,5,108] in symptomatic AD patients (in sporadic, ADAD, and DSAD). The response to treatment is, however, good, with 72–80% of patients without seizures after a year in monotherapy [4,5,8,58,108,109,110,111]. Despite this positive response to treatment in clinical series, there are no double-blind, placebo-controlled clinical trials to support the use of one ASM over another in AD-related epilepsy [112,113]. Levetiracetam and lamotrigine, broad-spectrum ASMs, are the most widely recommended drugs due to a better security profile compared to other ASMs [51,68,110,111,114,115,116,117]. Lacosamide and brivaracetam have been proposed as potential alternative ASMs in epilepsy in AD, but results supporting their use are still preliminary [76,114] (Table 2).

### 4.2. Impact of AD Treatments on Seizure Occurrence and Control

Table 3 summarizes the data on the effect of the most frequently used drug classes in symptomatic AD patients. There are no data to support neither to discontinue or not to initiate acetylcholinesterase inhibitors in subjects with AD and a personal history of seizures [118,119]. There is insufficient information to make a statement in the case of memantine [118]. Selective serotonin reuptake inhibitors or mirtazapine is preferred over other antidepressants. There are more data relating antipsychotics to worse seizure control [119], but if required, second-generation antipsychotics (especially quetiapine and risperidone) should be the first option in AD [118,119].

### 4.3. Antiseizure Medications (ASMs) as Possible Alzheimer’s Disease-Modifying Treatments

Neuronal hyperexcitability, synaptic dysfunction, IEDs, and electroclinical seizures are phenomena described from the preclinical stages of AD. As we have discussed, their presence, in turn, promotes the deposition and propagation of amyloid and tau proteins. ASMs, in addition to reducing the frequency of IEDs and electroclinical seizures, could also help reestablish the excitatory–inhibitory balance and normalize synaptic function, potentially positively influencing disease progression in AD.

#### 4.3.1. SV2A Ligands (Levetiracetam (LEV) and Brivaracetam (BVT))

Multiple potential benefits of LEV have been described in animal models, supporting its possible role as a modifying treatment for AD biology: (1) reduces glutamate release and glutamate-mediated excitotoxicity [6,58,119,120], favoring synaptic function recovery and reducing neuronal death [13]; (2) restores mitochondrial dysfunction [111,121]; (3) promotes neurogenesis and positively modifies hippocampal remodeling [111,121]; (4) suppresses neural hyperactivity in the hippocampus; and (5) decreases Aβ42 cortical levels and amyloid plaque burden [97]. Additionally, after the use of LEV, benefits have been reported in studies of transgenic animal models: improvement of learning [111] and memory deficits and spatial discrimination tasks [115] (Table 4). 

The promising results obtained in animal models have led to several clinical trials in humans proposing the use of LEV at low doses as a treatment to modify the clinical and biological course of AD. The most important findings in humans are as follows: (1) suppresses neural hyperactivity in the CA3 hippocampus region and dentate gyrus [13,23,51,122]; (2) normalizes the oscillation of rhythmic activity assessed by quantitative EEG [122,123]; and (3) improves cognitive functions globally assessed by MMSE and ADAS-Cog [13,51] and, especially, spatial memory and executive dysfunction in AD patients with IEDs [116,122]. Future clinical trial protocols for the use of LEV in AD aim to jointly evaluate the improvement of EEG abnormalities and clinical improvement (cognitive, behavioral, and functional) in relation to low-dose LEV administration [117].

Preliminary data from animal models suggest similar results to those with the use of BVT: (1) modifies the sensitivity of synaptic vesicles to calcium and reduces glutamate release and glutamate-mediated excitotoxicity [88]; (2) reduces the frequency of electroclinical seizures and IEDs [23,112]; and (3) improves memory dysfunction [23].

#### 4.3.2. Sodium Channel Blockers

In animal models, lamotrigine (LTG) has shown its potential to slow the biology and clinical progression of AD: (1) reduces the glutamate release from excitatory neurons [6,23,51,58,119]; (2) decreases the expression of BACE1 [124]; (3) inhibits mTOR signaling [124]; (4) attenuates selective CA1 hippocampal neuronal loss, upregulates antiapoptotic protein Bcl-2, and stimulates neurogenesis in the granule cell layer of dentate gyrus [125]; (5) reduces amyloid plaque density [69]; and (6) improves executive dysfunction [126]. In turn, there is anecdotal evidence in humans showing an improvement in naming and recognition tasks and depression scale scores in AD patients treated with low–moderate doses of LTG [23,51] (Table 4).

Lacosamide (LCS), which is a potential alternative for LTG, also has preliminary data from animal models showing: (1) the inhibition of Aβ-induced hyperphosphorylation of tau and (2) the inhibition of histone deacetylase, which regulates the expression of important genes for learning and memory processes, improving their dysfunctions [127].

Carbamazepine (CBZ), which is currently not a first- or second-choice treatment for epilepsy in AD, also has a beneficial effect in the biological progression of AD in animal models: (1) reduces glutamate-mediated excitotoxicity [125]; (2) increases noradrenergic tone [128]; and (3) decreases the burden of amyloid plaques [13].

#### 4.3.3. Calcium Channel Blockers

From animal models, there is information that suggests the capacity of gabapentin (GBP) to reduce (1) Aβ-induced toxicity [125,129] and (2) neuronal hyperexcitability in the context of AD [130] (Table 4).

#### 4.3.4. ASMs with Multiple Mechanisms

Valproic acid (VPA) is a broad-spectrum ASM currently not indicated in AD-related epilepsy, due to its suboptimal cognitive and motor security profile. It has, however, also shown evidence in animal models and cell cultures to: (1) reduce the amount of Aβ oligomers and neuritic plaques [69,131]; (2) inhibit the activity of GSK-3β and thus the amount of hyperphosphorylated tau [131,132]; (3) reduce neuronal loss activating the antiapoptotic protein Bcl-2 and promote neurogenesis via its histone deacetylase inhibitor capacity [133]; (4) stimulate GABAergic neuron differentiation; and (5) reduce glial differentiation [125,133,134]. Only one study including LOEU patients suggested the improvement of verbal fluency, but not other cognitive domains with the use of VPA [13] (Table 4).

Animal models have also suggested the potential biological benefit of topiramate (TPM) on AD progression, which: (1) stimulates GABA_A_ receptor function [114]; (2) blocks AMPA receptors; (3) acts as histone deacetylase; and (4) inhibits GSK-3β, reducing the amount of hyperphosphorylated tau [23]. Similar benefits for ZNS have also been suggested [125].

## 5. Conclusions

The excitatory–inhibitory imbalance in AD not only leads to an increased risk of epileptiform activity and electroclinical seizures, but also plays a key role in the progression of AD pathophysiology. ASMs could potentially not only ameliorate clinical and subclinical epileptiform activity, but also potentially modify the natural progression of the disease. Clinical trials to guide the use of ADM in AD-associated epilepsy and to evaluate the impact of ASM on A biomarkers and cognitive decline may represent a new therapeutic strategy to prevent and treat AD.

## Figures and Tables

**Figure 1 ijms-23-04307-f001:**
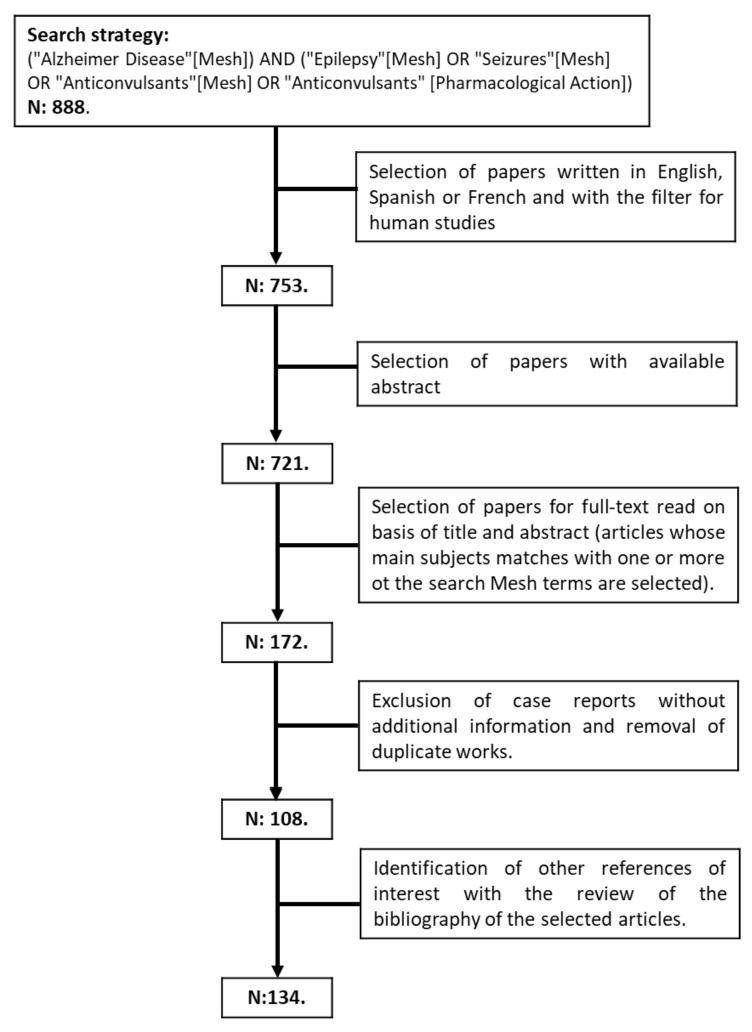
Flowchart of research strategy.

**Figure 2 ijms-23-04307-f002:**
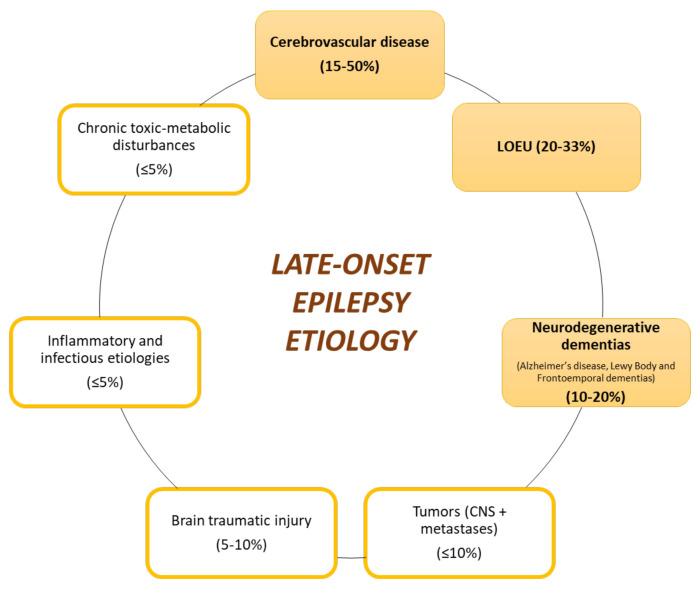
Identified causes of late-onset epilepsy. Cerebrovascular diseases, late-onset epilepsy of unknown etiology (LOEU), and neurodegenerative dementias are the most prevalent etiologies.

**Figure 3 ijms-23-04307-f003:**
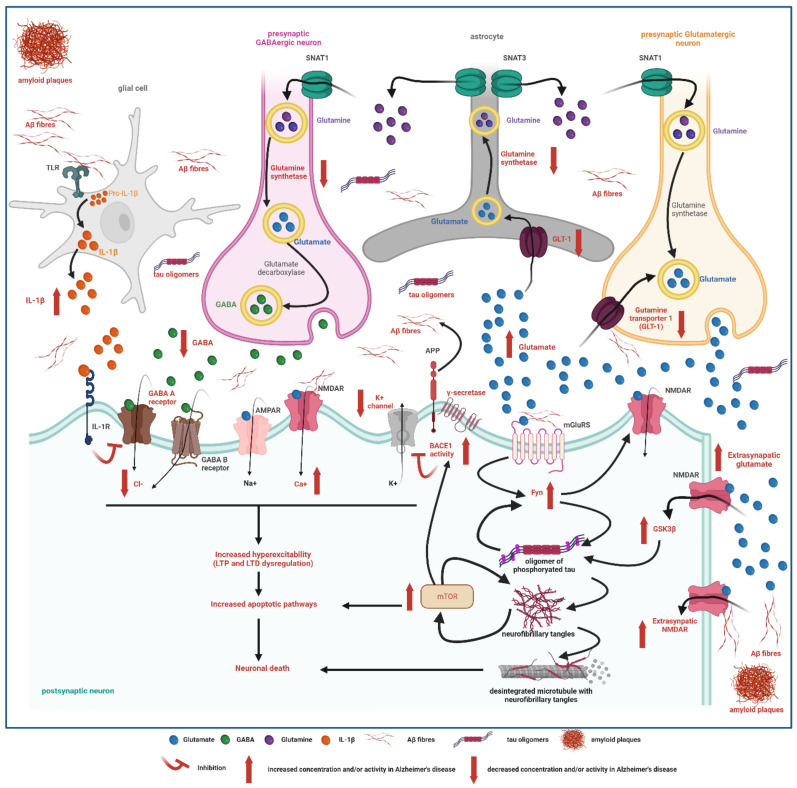
Possible mechanisms involved in the proexcitatory and proepileptic roles of soluble forms of amyloid (Aβ) and tau protein. LTP: long-term potentiation; LTD: long-term depression; TLR: Toll-like receptor; GSK3β: Glycogen synthase kinase 3 beta. Created with biorender.com (accessed on 1 January 2020).

**Table 1 ijms-23-04307-t001:** Summary of identified risk factors for development of epilepsy in the context of Alzheimer’s disease.

Suggested Risk Factors of Epilepsy in AD
**Sociodemographic:** Male sex. Younger age at the onset of symptoms (both in sporadic and autosomal dominant AD). **Clinical and anatomic features:** Longer disease duration (in years). Disease severity. Neuroimaging features: greater affectation of precuneus and atrophy pattern with parietal predominance. Chronic use of drugs that reduce seizure threshold (for instance, classic antipsychotics). **Comorbidities increasing the risk of epilepsy:** Cerebrovascular pathology (micro- and macrovascular damage, mainly with when there is cortical involvement). Brain traumatic injury.

**Table 2 ijms-23-04307-t002:** Recommendations for the use of antiseizure medications (ASMs) in Alzheimer’s disease (AD) based on scientific evidence and clinical practice experience. Na^+^: sodium, CBZ: carbamazepine, OXC: oxcarbazepine, ESL: eslicarbazepine, Ca^+^: calcium; NMDAR: N-Methyl-D-Aspartate receptor; AMPAR: α-amino-3-hydroxy-5-methyl-4-isoxazolepropionic acid receptor.

	SV2A Ligands		Na^+^ Channel Blockers			Multiple Mechanisms		Ca^+^ Channel Blockers	AMPAR Blocker
	Levetiracetam (LEV)	Brivaracetam (BVT)	Lamotrigine (LTG)	Lacosamide (LCS)	“Zepines” (CBZ, OXC, ESL)	Valproic Acid (VPA)	Zonisamide (ZNS) and Topiramate (TPM)	Pregabalin (PGB) and Gabapentin (GBP)	Perampanel (PER)
**Mechanism of action**	- Binds SV2A. - Blocks AMPA and NMDAR (reduces release of glutamate). - Induces GABA potentiation. - Effect on glycine or kainic-acid currents.	- Binds SV2A (20-fold higher affinity compared to LEV). -Minor block on NMDAR.	- Blocks voltage-dependent sodium channels.	- Blocks voltage-dependent sodium channels (enhancing slow inactivation).	- Blocks voltage-dependent sodium channels.	- GABA potentiation. - Blocks T-type calcium channels, sodium channels, and NMDAR.	- GABA potentiation (only TPM). - Blocks AMPAR (only TPM), T-type calcium channels (only ZNS), and voltage-dependent sodium channels.	- Blocks voltage-dependent calcium channels.	- AMPA glutamate receptor antagonist.
**Spectrum of efficacy**	- Broad-spectrum. Including antimyoclonic effect.	- Focal seizures. - Preclinical models: broad-spectrum efficacy.	- Broad-spectrum.	- Focal seizures.	- Focal seizures.	- Broad-spectrum.	- Broad-spectrum.	- Focal seizures.	- Focal seizures, generalized seizures (only as adjunctive therapy), useful for myoclonic seizures.
**Clinical experience in AD**	- First-line treatment. - Safety and absence of interactions.	- Well tolerated. - Less irritability than LEV. - Alternative for LEV or LTG.	- First-line treatment. - Less sedative and few cognitive adverse effects.	- Well tolerated. - Alternative for LEV or LTG.	- Not considered as first- or second-line treatment.	- Not considered as first- or second-line treatment.	- Not considered as first- or second-line treatment.	- Not considered as first- or second-line treatment.	- Possible alternative treatment, study data are lacking. - No data on cognitive side effects.
**Potential limitations and risks in AD**	- Dose-dependent somnolence and irritability. - 10–15% stop due to neuropsychiatric side effects.	- Irritability but with lower frequency compared to LEV.	- Unsteadiness. - Onset insomnia. - May exacerbate myoclonic seizures.	- Unsteadiness (less frequent than others Na+ blockers). - May exacerbate myoclonic seizures.	- Cognitive impairment related with decreased cholinergic tone (less frequent with ESL). - Unsteadiness.	- Encephalopathy, hyperammonemia. - May induce cognitive impairment and/or motor worsening (tremor).	- Cognitive adverse effects (less frequent with ZNS).	- Less effective. - Cognitive slowing. - Dizziness.	- Dizziness. - Aggression and hostility (special caution if neuropsychiatric symptoms with LEV).

**Table 3 ijms-23-04307-t003:** Impact on seizure threshold of frequently used symptomatic treatments in AD. ^a.^ Low to moderate impact, ^b.^ moderate impact, ^c.^ both anti- and proepileptic effects reported.

**Neutral.**	Acetylcholinesterase inhibitors.
	Antidepressants: Selective serotonin reuptake inhibitors.
	Antipsychotics: Quetiapine and risperidone.
**Decrease seizure threshold.**	Antidepressants ^a^*:* Tricyclic antidepressants and bupropion.
	Antipsychotics ^b^: Clozapine, chlorpromazine and haloperidol.
**Controversy.**	Memantine ^c^

**Table 4 ijms-23-04307-t004:** Summary of potential clinical and biological benefits as AD disease-modifying treatments of different ASMs both from animal models and cell cultures and humans. We reviewed published and ongoing studies to analyze the potential benefits of this intervention. At present, most data are obtained from pre-clinical models. The most promising molecule is LEV. CBZ: carbamazepine, OXC: oxcarbazepine, ESL: eslicarbazepine acetate, fMRI: functional magnetic resonance imaging; EEG: electroencephalogram; E-I system: excitatory–inhibitory system; GLUT: glutamate, GABA: γ-aminobutyric acid, BACE 1: beta-site amyloid precursor protein cleaving enzyme 1, HDAC: histone deacetylase, BCL2: B-cell lymphoma 2, LOEU: late-onset epilepsy of unknown etiology, GSK3β: Glycogen synthase kinase 3, p-tau: Phosphorylated tau; NA tone: noradrenergic tone, Aβ: β-amyloid.

		SV2A Ligands		Na^+^ Channel Blockers			Multiple Mechanisms		Ca^+^ Channel Blockers
		Levetiracetam (LEV)	Brivaracetam (BVT)	Lamotrigine (LTG)	Lacosamide (LCS)	“Zepines” (CBZ, OXC, ESL)	Valproic Acid (VPA)	Zonisamide (ZNS) and Topiramate (TPM)	Pregabalin (PGB) and Gabapentin (GBP)
**HUMAN MODELS**		- Improve attention, verbal fluency, visuospatial functions, and hippocampal-related memory tasks. - Reduce hippocampal hyperactivity (assessed by fMRI and EEG).	- Expected to be similar to LEV.	- Better performance in naming and recognition tasks. - Improvement of affective symptoms (mainly depression).			- Single study in LOEU: improve verbal fluency but no other cognitive domains.		
**ANIMAL MODELS AND CELL CULTURES**	**Fibrillar and amyloid plaque deposition**	- ↓ Aβ42 oligomers and fibrils, and amyloid plaque burden.		- ↓ BACE1 (via ↓mTOR): ↓ amyloid plaque density		- ↓ Aβ plaques	- ↓ Aβ oligomers and formation of neuritic plaques.		- Neuro-protection: interfere with Aβ-induced toxicity.
	**Tau deposition and/or hyperphosphorylation**				- ↓ Aβ-induced hyperphosphorylation of tau.		- ↓ GSK3β activity: ↓ p-tau.	- ↓ GSK3β: ↓ p-tau.	
	**Neurogenesis and/or hippocampal remodeling**	- Modify positively hippocampal remodeling. - Restore neurogenesis.		- ↓ CA1 hippocampal neuronal loss. - ↓ HDAC, ↑BCL2: neurogenesis in the granule cell layer of dentate gyrus.	- ↓ HDAC activity.		- ↑ bcl-2:↓ apoptosis. - ↑ Neuronal progenitor proliferation by ↑ cyclin D2.	- ↓ HDAC activity.	
	**Others**	- Repair mitochondrial dysfunction. - Modify the excitotoxicity mediated by GLUT. - Improve synaptic function. - ↓ hippocampal hyperexcitability.	- Normalize the E-I system imbalance. - Modify sensitivity of synaptic vesicles to Ca^+^: reduce release of NT (GLUT and GABA) in hippocampus.	- ↓ Neuroinflammation - ↓ GLUT release.		- ↓ GLUT-mediated excitatory signaling. - ↑ NA tone.	- ↑ GABAergic neuron differentiation. - ↓Neuroinflammation.	- ↑ GABAergic tone.	- ↓ neuronal hyperexcitability.
	**Cognitive function improvement**	- Improve learning and memory deficits and spatial discrimination tasks.	- Enhance performance in memory tasks.	- Ameliorate executive dysfunction.	- May improve disrupted memory.				

## Data Availability

Not applicable.

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
