# Peer review of "Mechanisms Involved in Epileptogenesis in Alzheimer’s Disease and Their Therapeutic Implications"

_ijms, 2022, doi:10.3390/ijms23084307_

Round 1

Reviewer 1 Report

I think this is a very clear review of the epileptogenicity of AD and ASM treatment. This  paper provides a good overview of the relationship between dementia and epilepsy, and the current state of research in this important chronic neurological disease in the elderly.

To further improve it, please consider some of the points listed below.

  1. The authors selected “anticonvulsants” as one of the key words in their paper review. However, I think more frequently used keywords were “antiepileptic drugs” or “antiseizure medication”. Please explain why “anticonvulsants” was chosen. If the authors have a data using the other two keywords, what difference would occur in their review?
  2. Table 2 is too elementary and seems odd. The table is as far from the high content of this review. I think the authors as well as readers don't need this Table.
  3. I think a few things need to be modified in Figure 3: 
    1. The embedded cartoon on the top right that explained the interaction of glial cell supposed to represent the release of IL-1β from the inflammatory process via TLRs. However, I think it could be drawn them together in a large cartoon. 
    2. In the main cartoon, the pink dots are probably shown IL-1β, the blue dots shows glutamate, and the green dots are GABA, but I think it is very confusing. The authors had better to explain those in somewhere the margin.
    3. I cannot understand the meaning of the red arrow. 
  4. Table 4 is difficult to understand. confusing, I think it would be easier to understand if “Neutral”, “Decreasing seizure threshold” and “Controversy” are written separately.
  5. Table 5 is also confusing in the following ways: 
    1. The meanings whether "+" and "-" are made the readers confused. I confused those “favorable” or “unfavorable” for epilepsy. 
    2. In Table 3, drugs are listed in rows and the mechanisms are written in columns. I think it would be easier for readers to understand if you wrote as the same manner. 
    3. Table 3 and Table 5 should be concise and may integrate together.
  6. In the EEG terminology of American Clinical Neurophysiology, “triphasic waves” has been changed to “continuous 2/s generalized periodic discharges (with triphasic morphology) “(Clin Neurophysiol 2013; 30; 1-27). It may have been “tripharic waves “ in the original reviewed paper, but I think it would be better to use a current standard EEG terminology. 

Author Response

I think this is a very clear review of the epileptogenicity of AD and ASM treatment. This  paper provides a good overview of the relationship between dementia and epilepsy, and the current state of research in this important chronic neurological disease in the elderly.

To further improve it, please consider some of the points listed below.

  1. The authors selected “anticonvulsants” as one of the key words in their paper review. However, I think more frequently used keywords were “antiepileptic drugs” or “antiseizure medication”. Please explain why “anticonvulsants” was chosen. If the authors have a data using the other two keywords, what difference would occur in their review?

Thank you for your comment. We explain the reason for the choice of anticonvulsants, which we share that it is a term in disuse today. The reason is that when searching using Mesh terms there is no Mesh term “antiseizure drugs” or “antiseizure medications” nor “antiepileptic drugs”. On the other hand, the term Mesh “anticonvulsants” (ID in Pubmed 000927) includes all of the following sections that we believe are necessary for the review we wished to perform:

  • Anticonvulsive Agent
  • Agent, Anticonvulsive
  • Anticonvulsive Drug
  • Drug, Anticonvulsive
  • Anticonvulsive Drugs
  • Drugs, Anticonvulsive
  • Anticonvulsant
  • Anticonvulsant Drugs
  • Drugs, Anticonvulsant
  • Anticonvulsive Agents
  • Agents, Anticonvulsive
  • Anticonvulsant Drug
  • Drug, Anticonvulsant
  • Antiepileptic Agents
  • Agents, Antiepileptic
  • Antiepileptics
  • Antiepileptic Drug
  • Drug, Antiepileptic
  • Antiepileptic
  • Antiepileptic Agent
  • Agent, Antiepileptic
  • Antiepileptic Drugs
  • Drugs, Antiepileptic
  • 1,3-ditolylguanidine (Supplementary Concept)
  • 1-(4-chlorophenyl)-4-piperidin-1-yl-1,5-dihydroimidazol-2-one (Supplementary Concept)
  • 2,3-dioxo-6-nitro-7-sulfamoylbenzo(f)quinoxaline (Supplementary Concept)
  • 2,3-piperidinedicarboxylic acid (Supplementary Concept)
  • 2-(2,3-dicarboxycyclopropyl)glycine (Supplementary Concept)
  • 2-amino-4-methyl-5-phosphono-3-pentenoic acid (Supplementary Concept)
  • 2-amino-4-phosphonobutyric acid (Supplementary Concept)
  • 2-amino-7-phosphonoheptanoic acid (Supplementary Concept)
  • 2-fluoro-2-phenyl-1,3-propanediyl dicarbamate (Supplementary Concept)
  • 2-propyl-2-pentenoic acid (Supplementary Concept)
  • 3-(2-carboxypiperazin-4-yl)propyl-1-phosphonic acid (Supplementary Concept)
  • 4-amino-3-phenylbutyric acid (Supplementary Concept)
  • 4-phenyl-perhydropyrrole(1,2-a)pyrazine-1,3-dione (Supplementary Concept)
  • 5-(2-cyclohexylidene-ethyl)-5-ethylbarbiturate (Supplementary Concept)
  • 6-(1H-imidazol-1-yl)-7-nitro-2,3(1H,4H)-quinoxalinedione (Supplementary Concept)
  • 6-methoxytryptoline (Supplementary Concept)
  • 7-nitroindazole (Supplementary Concept)
  • abecarnil (Supplementary Concept)
  • Acetazolamide (MeSH Term)
  • alpha-hexachlorocyclohexane (Supplementary Concept)
  • angelicin (Supplementary Concept)
  • anthranilic acid (Supplementary Concept)
  • bemethyl (Supplementary Concept)
  • benzobarbital (Supplementary Concept)
  • bretazenil (Supplementary Concept)
  • brivaracetam (Supplementary Concept)
  • Bromides (MeSH Term)
  • Cannabidiol (MeSH Term)
  • Carbamazepine (MeSH Term)
  • Cenobamate (Supplementary Concept)
  • CGP 39551 (Supplementary Concept)
  • chlordesmethyldiazepam (Supplementary Concept)
  • Chlormethiazole (MeSH Term)
  • Clobazam (MeSH Term)
  • Clonazepam (MeSH Term)
  • Clorazepate Dipotassium (MeSH Term)
  • denzimol (Supplementary Concept)
  • deramciclane (Supplementary Concept)
  • Diazepam (MeSH Term)
  • Dimethadione (MeSH Term)
  • dipropylacetamide (Supplementary Concept)
  • DN 1417 (Supplementary Concept)
  • doramectin (Supplementary Concept)
  • eperisone (Supplementary Concept)
  • eslicarbazepine acetate (Supplementary Concept)
  • Estazolam (MeSH Term)
  • Ethosuximide (MeSH Term)
  • ethotoin (Supplementary Concept)
  • ethylphenylhydantoin (Supplementary Concept)
  • ezogabine (Supplementary Concept)
  • Felbamate (MeSH Term)
  • fludiazepam (Supplementary Concept)
  • Flunarizine (MeSH Term)
  • fosphenytoin (Supplementary Concept)
  • Gabapentin (MeSH Term)
  • gaboxadol (Supplementary Concept)
  • gidazepam (Supplementary Concept)
  • glutamic acid diethyl ester (Supplementary Concept)
  • GYKI 52466 (Supplementary Concept)
  • indeloxazine (Supplementary Concept)
  • indol-3-yl pyruvic acid (Supplementary Concept)
  • kavain (Supplementary Concept)
  • L 701324 (Supplementary Concept)
  • Lacosamide (MeSH Term)
  • Lamotrigine (MeSH Term)
  • Levetiracetam (MeSH Term)
  • Lorazepam (MeSH Term)
  • loreclezole (Supplementary Concept)
  • Magnesium Sulfate (MeSH Term)
  • mebeverine (Supplementary Concept)
  • Medazepam (MeSH Term)
  • Mephenytoin (MeSH Term)
  • Mephobarbital (MeSH Term)
  • Meprobamate (MeSH Term)
  • methsuximide (Supplementary Concept)
  • milacemide (Supplementary Concept)
  • N-(4,4-diphenyl-3-butenyl)nipecotic acid (Supplementary Concept)
  • N-desmethylclobazam (Supplementary Concept)
  • NCS 382 (Supplementary Concept)
  • neo-kyotorphin (Supplementary Concept)
  • neurotropin (Supplementary Concept)
  • nimetazepam (Supplementary Concept)
  • Nitrazepam (MeSH Term)
  • NNC 711 (Supplementary Concept)
  • Org 2766 (Supplementary Concept)
  • Oxcarbazepine (MeSH Term)
  • padsevonil (Supplementary Concept)
  • Paraldehyde (MeSH Term)
  • PD 117302 (Supplementary Concept)
  • phenazepam (Supplementary Concept)
  • pheneturide (Supplementary Concept)
  • Phenobarbital (MeSH Term)
  • Phenytoin (MeSH Term)
  • pipequaline (Supplementary Concept)
  • Pregabalin (MeSH Term)
  • Primidone (MeSH Term)
  • progabide (Supplementary Concept)
  • progabide acid (Supplementary Concept)
  • remacemide (Supplementary Concept)
  • Riluzole (MeSH Term)
  • rimcazole (Supplementary Concept)
  • rufinamide (Supplementary Concept)
  • ryodipine (Supplementary Concept)
  • sidnocarb (Supplementary Concept)
  • stiripentol (Supplementary Concept)
  • sulthiame (Supplementary Concept)
  • taglutimide (Supplementary Concept)
  • Thiopental (MeSH Term)
  • thioperamide (Supplementary Concept)
  • Tiagabine (MeSH Term)
  • Tiletamine (MeSH Term)
  • tizanidine (Supplementary Concept)
  • Topiramate (MeSH Term)
  • tramiprosate (Supplementary Concept)
  • Trimethadione (MeSH Term)
  • U 54494A (Supplementary Concept)
  • Valproic Acid (MeSH Term)
  • vanillin (Supplementary Concept)
  • Vigabatrin (MeSH Term)
  • zaleplon (Supplementary Concept)
  • ZK 91296 (Supplementary Concept)
  • ZK 93423 (Supplementary Concept)
  • ZK 93426 (Supplementary Concept)
  • Zonisamide (MeSH Term)

We must admit that performing a search without using the Mesh term "anticonvusants" and instead using terms such as antiseizure medications or antiseizure drugs or antiepileptic drugs would have increased the number of articles initially found. But we believe (we do not have the data) that the search would have been less efficient as we would probably have discarded more articles for further critical reading after reviewing the title and abstract. Searching by Mesh terms would have precisely that purpose at least at the more theoretical level. On the other hand, we believe that by adding not only articles identified by the initial search with the Mesh term "anticonvulsants" but also those referenced in these articles, we have been able to minimize the possible error in the selection.

  1. Table 2 is too elementary and seems odd. The table is as far from the high content of this review. I think the authors as well as readers don't need this Table.

We were aware that it was a very simple table but we believed that it could be useful for readers in the health care field and could be complementary to the text written in that section. Nevertheless, and fully understanding the constructive criticism made, we have decided to remove it from the revision.

  1. I think a few things need to be modified in Figure 3: 
    1. The embedded cartoon on the top right that explained the interaction of glial cell supposed to represent the release of IL-1β from the inflammatory process via TLRs. However, I think it could be drawn them together in a large cartoon. 
    2. In the main cartoon, the pink dots are probably shown IL-1β, the blue dots shows glutamate, and the green dots are GABA, but I think it is very confusing. The authors had better to explain those in somewhere the margin.
    3. I cannot understand the meaning of the red arrow. 

We have modified the figure. We believe that we have included the glial cell/IL-1B participation in a better way. We have added information about the meaning of the colours and we have explained the significance of the red arrows.

  1. Table 4 is difficult to understand. confusing, I think it would be easier to understand if “Neutral”, “Decreasing seizure threshold” and “Controversy” are written separately.

We have modified table 4 to try to make easier to understand. Thank you four your suggestion.

  1. Table 5 is also confusing in the following ways: 
    1. The meanings whether "+" and "-" are made the readers confused. I confused those “favorable” or “unfavorable” for epilepsy. 
    2. In Table 3, drugs are listed in rows and the mechanisms are written in columns. I think it would be easier for readers to understand if you wrote as the same manner. 
    3. Table 3 and Table 5 should be concise and may integrate together.

We have slightly modified the structure of the previous table 3 and 5. We hope that now it’s easier to read each of them. We have created a single table combining table 3 and Table 5 what we finally have decided to remove it. We consider that it could be useful to separate information about current clinical practice from preliminary evidence of the utility of antiseizure medications as potential disease modifying treatments in Alzheimer’s disease. We believe that both tables have different utilities: 1) the lack of clinical trials supporting the use of one antiseizure medication over another we think makes it advisable to have a table that collects the information for clinical utility (recommendations based on reading the articles and the authors' experience in clinical practice) and 2) to distinguish in detail the available evidence both in animal models and preliminary evidence in humans of these antiseizure medications as disease-modifying treatments for Alzheimer's disease we think it can be interesting for readers, and if it is not included in the table we believe that it should be included in the text of the article, which would probably make it less attractive to some readers.

  1. In the EEG terminology of American Clinical Neurophysiology, “triphasic waves” has been changed to “continuous 2/s generalized periodic discharges (with triphasic morphology) “(Clin Neurophysiol 2013; 30; 1-27). It may have been “tripharic waves “ in the original reviewed paper, but I think it would be better to use a current standard EEG terminology. 

Thank you for your suggestion. We have included it.

Reviewer 2 Report

Reading and studying the manuscript of Altuna et al. it was very interesting. In the paper, the authors aim to highlight correlations between the two diseases (epilepsy and Alzheimer's Dementia). In the paper, the authors reviewed the mechanisms involved in both epileptogenesis and A.D., showing that patients with epilepsy have an increased risk of developing Alzheimer's Dementia. Therapeutic options for treating epileptic seizures in patients with A.D. were also reviewed, as well as other symptoms present in A.D.

The article is written in a concise and orderly manner, respecting the structure of the journal. The bibliographic references are in acceptable numbers, some of them being as recent studies as possible. I appreciate the fact that the authors included tables and images in which they synthesized the risk factors involved, the causes of epilepsy, numerous epileptogenic mechanisms involved in A.D. and the impact of therapy on various symptoms.

However, I have a few comments:

  1. In the section 3.4.1. Neuroinflammation, the authors should discuss the role of the blood-brain barrier and how the alteration of this barrier can be influenced by the pathological processes in A.D.
  2. In the section 3.4.2. mTOR's role in A.D. should be detailed.
  3. The quality of image from Figure no. 3. should be improved. Also, the font size should be increased for better readability. Furthermore, various mechanisms are also mentioned in the image legend but no explanation is given for these mechanisms.
  4. In the section 4.3.1., in this sentence: “Also in animal models, clinical benefits in AD with the use of LEV have been reported: improvement of learning and memory deficits and spatial discrimination tasks”, I think that these are preclinical trials, not clinical trials, being made on animal models.
  5. The English language, although quite good, should be improved.
  6. While reading this manuscript I noticed that there was some overlap with other previous studies (some plagiarism concerns). At a simple check with specialized software, the percentage of similarity is higher than the usual limit to be published. Some examples:

Finally, these aggregates have been found in the resection material of refractory temporal lobe epilepsies suggesting that epilepsy leads to amyloid and tau aggregates. Some epileptic syndromes, such as medial temporal lobe epilepsy share structural and functional neuroimaging findings with AD, leading to over-lapping symptomatology such as episodic memory deficits and toxic synergistic effects.

In this respect, the existence of epileptiform activity and electroclinical seizures in AD appears to accelerate progression of cognitive decline and the presence of cognitive decline is much more prevalent in epileptic patients than in elderly without epilepsy. Not-withstanding their clinical significance, the diagnosis of clinical seizures in AD is a challenge.

 Most are focal and manifest with altered level of consciousness without motor symptoms, and are often interpreted as cognitive fluctuations. Finally, despite the frequent association of epilepsy and AD dementia, there is a lack of clinical trials to guide the use of antiseizure medications (ASM). There is also a potential role for ASMs to be used as disease-modifying drugs in AD. Keywords: ability. 1.

In addition AD and epilepsy share several risk factors including advanced age, cardiovascular risk factors and damage to the cerebral vasculature, history of brain traumatic injury and the presence of the 4 allele of the APOE gene [17,21]. Importantly, both amyloid and tau protein aggregates have proepileptic effects [9,22 24] and conversely, these defining anatomopathologic hallmarks of AD have been described in surgical material from refractory mesial temporal lobe epilepsies [25]. There are also shared pathophysiological processes in the two diseases.

Earlier onset of symptoms (both in sporadic and autosomal dominant AD). Clinical and anatomic features: Longer disease duration (in years). Disease severity. Greater involvement of precuneus and a parietal-dominant atrophy pattern. Chronic use of antipsychotics. 4 Comorbidities increasing the risk of epilepsy: Cerebrovascular pathology (mainly with when there is cortical involvement). Brain traumatic injury.

There are many other sentences copied directly from other publications. Thus, I have stopped my review here and I suggest that you first remove the similarities from your manuscript and resubmit the paper.

Author Response

Reading and studying the manuscript of Altuna et al. it was very interesting. In the paper, the authors aim to highlight correlations between the two diseases (epilepsy and Alzheimer's Dementia). In the paper, the authors reviewed the mechanisms involved in both epileptogenesis and A.D., showing that patients with epilepsy have an increased risk of developing Alzheimer's Dementia. Therapeutic options for treating epileptic seizures in patients with A.D. were also reviewed, as well as other symptoms present in A.D.

Thank your for your kind words for your suggestions in order to improve our work.

The article is written in a concise and orderly manner, respecting the structure of the journal. The bibliographic references are in acceptable numbers, some of them being as recent studies as possible. I appreciate the fact that the authors included tables and images in which they synthesized the risk factors involved, the causes of epilepsy, numerous epileptogenic mechanisms involved in A.D. and the impact of therapy on various symptoms.

However, I have a few comments:

  1. In the section 3.4.1. Neuroinflammation, the authors should discuss the role of the blood-brain barrier and how the alteration of this barrier can be influenced by the pathological processes in A.D.

We have added a comment about the dysfunction of BBB in AD and TLE, and about its suggested proexcitatory/proepileptic effect.

  1. In the section 3.4.2. mTOR's role in A.D. should be detailed.

We have added more information about this topic as suggested.

  1. The quality of image from Figure no. 3. should be improved. Also, the font size should be increased for better readability. Furthermore, various mechanisms are also

mentioned in the image legend but no explanation is given for these mechanisms.

     We have slightly modified the Figure number 3. We hope that quality is better now. We have significantly increased the font size as suggested. We have added information in the main manuscript about the mechanisms mentioned in the figure.

  1. In the section 4.3.1., in this sentence: “Also in animal models, clinical benefits in AD with the use of LEV have been reported: improvement of learning and memory deficits and spatial discrimination tasks”, I think that these are preclinical trials, not clinical trials, being made on animal models.
  2. The English language, although quite good, should be improved.

We have sent the manuscript to English correction section of the Journal in order to improve its quality.

  1. While reading this manuscript I noticed that there was some overlap with other previous studies (some plagiarism concerns). At a simple check with specialized software, the percentage of similarity is higher than the usual limit to be published. Some examples:

As we were surprised by this statement and the extracts copied from the texts (we were aware that we had not copied them), we have requested that at the same time as the English correction, a control of identification of possible plagiarism be carried out. This identified a similarity of 1 or less than 1% with all the references and only very high (up to 80%) with the previous version of this same work (same title, same authors, it’s easy to check) that is currently available on the preprint website. Therefore, we do not believe there is any possible plagiarism.

Finally, these aggregates have been found in the resection material of refractory temporal lobe epilepsies suggesting that epilepsy leads to amyloid and tau aggregates. Some epileptic syndromes, such as medial temporal lobe epilepsy share structural and functional neuroimaging findings with AD, leading to over-lapping symptomatology such as episodic memory deficits and toxic synergistic effects.

In this respect, the existence of epileptiform activity and electroclinical seizures in AD appears to accelerate progression of cognitive decline and the presence of cognitive decline is much more prevalent in epileptic patients than in elderly without epilepsy. Not-withstanding their clinical significance, the diagnosis of clinical seizures in AD is a challenge.

 Most are focal and manifest with altered level of consciousness without motor symptoms, and are often interpreted as cognitive fluctuations. Finally, despite the frequent association of epilepsy and AD dementia, there is a lack of clinical trials to guide the use of antiseizure medications (ASM). There is also a potential role for ASMs to be used as disease-modifying drugs in AD. Keywords: ability. 1.

In addition AD and epilepsy share several risk factors including advanced age, cardiovascular risk factors and damage to the cerebral vasculature, history of brain traumatic injury and the presence of the 4 allele of the APOE gene [17,21]. Importantly, both amyloid and tau protein aggregates have proepileptic effects [9,22 24] and conversely, these defining anatomopathologic hallmarks of AD have been described in surgical material from refractory mesial temporal lobe epilepsies [25]. There are also shared pathophysiological processes in the two diseases.

Earlier onset of symptoms (both in sporadic and autosomal dominant AD). Clinical and anatomic features: Longer disease duration (in years). Disease severity. Greater involvement of precuneus and a parietal-dominant atrophy pattern. Chronic use of antipsychotics. 4 Comorbidities increasing the risk of epilepsy: Cerebrovascular pathology (mainly with when there is cortical involvement). Brain traumatic injury.

There are many other sentences copied directly from other publications. Thus, I have stopped my review here and I suggest that you first remove the similarities from your manuscript and resubmit the paper.

Round 2

Reviewer 2 Report

The authors made the changes in the article as requested which led to a significant improvement in the scientific quality of the paper.

However, I did not see any changes to the paragraph in section 4.3.1. “Also in animal models, clinical benefits in AD with the use of LEV have been reported: improvement of learning and memory deficits and spatial discrimination tasks”.

My proposal is to change the above paragraph as follows: "Additionally, after the use of LEV, benefits have been reported in studies of transgenic animal models and in AD patients: improvement of learning and memory deficits and spatial discrimination tasks."

In my opinion, the paper may be considered for publication.

Author Response

The authors made the changes in the article as requested which led to a significant improvement in the scientific quality of the paper.

Thank you for your kind words.

However, I did not see any changes to the paragraph in section 4.3.1. “Also in animal models, clinical benefits in AD with the use of LEV have been reported: improvement of learning and memory deficits and spatial discrimination tasks”.

My proposal is to change the above paragraph as follows: "Additionally, after the use of LEV, benefits have been reported in studies of transgenic animal models and in AD patients: improvement of learning and memory deficits and spatial discrimination tasks."

Thank you again for your kind suggestions. We have added your phrase and removed ours (we have not included only the AD patients concept because we facilitate this information in the next paragraph). But if you consider necessary to repeat the same idea we would incorporate it.

Changes are emphasized in the manuscript using the red colour.

Round 3

Reviewer 2 Report

In my opinion, the paper may be considered for publication.